# Manufacturing of Mg-Ti Couples at Different Heat Treatment Temperatures and Their Corrosion Behavior in Chloride Solutions

**DOI:** 10.3390/ma12081300

**Published:** 2019-04-20

**Authors:** El-Sayed M. Sherif, Abdulaziz N. AlHazaa, Hany S. Abdo

**Affiliations:** 1Research Chair for Tribology, Surface, and Interface Sciences (TSIS), Department of Physics and Astronomy, College of Science, King Saud University, P.O. Box 2455, Riyadh 11451, Saudi Arabia; aalhazaa@ksu.edu.sa; 2Center of Excellence for Research in Engineering Materials (CEREM), King Saud University, P.O. Box 800, Al-Riyadh 11421, Saudi Arabia; habdo@ksu.edu.sa; 3Electrochemistry and Corrosion Laboratory, Department of Physical Chemistry, National Research Centre, El-Buhouth St., Dokki, Cairo 12622, Egypt; 4King Abdullah Institute for Nanotechnology (KAIN), King Saud University, P.O. Box 2455, Riyadh 11451, Saudi Arabia; 5Mechanical Design and Materials Department, Faculty of Energy Engineering, Aswan University, Aswan 81521, Egypt

**Keywords:** AZ31 alloy, heat treatment, Mg-Ti couple, corrosion, potentiodynamic polarization, surface morphology

## Abstract

In this study, rods of magnesium alloy and titanium alloy were cut to have similar height of about 5mm and size of 10 mm × 10 mm to fabricate three Mg-Ti couples. The Mg-Ti couple was heat treated at 540 °C, 570 °C, and 600 °C. The corrosion of these couples have been investigated and compared with AZ31 alloy. Potentiodynamic polarization and electrochemical impedance spectroscopy measurements were employed to study the corrosion behavior after 1.0 h and 48 h exposure to 3.5% NaCl solutions. The morphology of surfaces was examined by scanning electron microscopy (SEM) and the profile analysis was collected using an energy dispersive X-ray (EDX) analyzer after 5 days immersion in the chloride solutions. It is found that coupling Mg with Ti reduces the corrosion of AZ31 alloy, which further decreased with the increase of the temperature of treatment. Prolonging the time of exposure from 1.0 h to 48 h remarkably decreased the corrosion of the couples as well.

## 1. Introduction

Magnesium (Mg) and Mg alloys have various applications because of their density, damping and heat dissipation, and excellent electromagnetic shield [1,2,3,4,5,6,7,8]. The surfaces of these alloys usually develop oxide film with severe corrosion particularly in chloride containing environments. Such an aggressive ion, Cl^−^ ion, penetrates the oxide film attacking the alloy surface and reacting with the metal substrate leading to its corrosion. AZ31 Mg alloy is well known to be one of the most applicable alloys in the industrial sector [9]. This Mg alloy has high strength, low density, good ductility and excellent machinability and weldability at room temperature [10,11,12]. For that it is being used in aircraft fuselages, vehicles, cell phone and laptop cases, concrete tools, and speaker cones [12,13,14,15]. AZ31 alloy has both Al and Zn; the increase of Al content increases the tensile strength, yield strength, resistance against corrosion, and oxidation resistance. This effect affects the mechanical properties, where it decreases the castability and weldability of AZ31 alloy [15]. Also, the presence of Zn within the AZ31 alloy increases its castability and corrosion resistance [10]. 

AZ31 alloy and Ti-6Al-4V alloy are employed in the industry of automobiles because of their terrific mechanical and physical properties [16,17]. Where and amongst all structural metals, magnesium has the best strength-to-weight ratio [18]. Due to its mild weight, magnesium has been an appealing preference for automobile and aerospace industries [18,19]. While, the strength-to-weight ratio and the resistance to corrosion are also high for titanium; for this reason it is far an appealing choice for aerospace industries. However, the value of titanium in comparison to other metals (i.e. aluminum and magnesium) limits its use. Therefore, systems constructed using magnesium, titanium, aluminum, and their combinations is expected to have a growing impact on various industries specifically in aerospace and automotive industries with a purpose to reduce fuel consumption, greenhouse gases, and enhance performance of energy saving systems. Moreover, it has been reported that it is important for magnesium alloys to be joined with more noble metals to find more applications for these materials in industry [13].

The corrosion resistance of the magnesium alloys is important to be considered when joining Mg alloys because Mg and its alloys are reactive with lower resistance against corrosion in harsh environments. The occurrence and severity of galvanic corrosion between alloys such as our tested materials depend on some factors [12,20,21,22] such as the potential difference between those metals or alloys; the higher the potential difference the more severe the corrosion is. The metal with more negative potential is the anode and the cathode is the nobler metal. The anode/cathode area ratio and the distance between anode and cathode also affect the severity of the galvanic corrosion [20,21]. The corrosion intensity for galvanic decreases when there is a long distance and less potential difference between the anode and cathode materials. For galvanic corrosion to take place there must be a conductive path between the anode and cathode. This path allows the metal ions to move from the more anodic metal to the more cathodic one inside the solution. Also, an electrical path is necessary for the galvanic corrosion to take place between the two coupled metals. In all cases, the anode material dissolves to protect the cathode.

In the current work, magnesium AZ31 and titanium Ti-6Al-4V alloys were coupled together using thin filler material contains 10 µm Cu and 50 µm Sn. The combination of Cu and Sn at the interface were used to enhance the diffusion process. Magnesium and titanium samples were bonded under 1 MPa uniaxial pressures in period of time of 20 min. The bonding temperature was selected as variable bonding parameters. Three bonded were successfully achieved at 540 °C, 570 °C, and 600 °C. It was reported [23,24,25] that the alloying elements, the grain size, and the microstructure affect the corrosion performance of Mg and its alloys in NaCl solutions. As example, Mg alloys with low amounts of Cu, Fe, and Ni will have high corrosion resistance. One aspect to look at when joining dissimilar alloys is the galvanic corrosion properties because there is a large potential difference between titanium and magnesium. Therefore, this study investigates the effect of bonding temperature on the galvanic corrosion of the coupled Mg-Ti in one of the most corrosive media, sodium chloride solutions.

## 2. Materials and Methods

### 2.1. Fabrication of Magnesium-Titanium Couples

The rods of magnesium alloy and titanium alloy were cut with similar sample height of 5 mm and size of 10 mm × 10 mm. The surfaces of the couples were ground using SiC paper to 1000 grit finish, after which they were cleaned by ethanol then washed by distilled water and finally dried and kept inside desiccators.

### 2.2. Coating Cu on Mg and Ti Alloy 

The Mg and Ti rods were inserted inside the chamber of a thermal evaporation system called Leybold coating system UNIVEX 300 (Cologne, Germany) with thickness monitoring as depicted in Figure 1. A pure Cu wire 99.99% was positioned on a resistive tungsten basket. A vacuum level of 10^−5^ mbar was reached and the current through the resistive tungsten basket was increased till evaporation of Cu took place. The evaporation process was continued for two minutes for obtaining a thickness of 5 µm coating on the surface of the Mg and Ti alloys.

### 2.3. Coupling Mg-Ti Samples

After coating the surfaces by Cu, Sn interlayer with thickness of 50 µm was inserted between each Ti and Mg couple. Bonding the samples of the Mg-Ti was done using high frequency heat induction furnace (HFHIF). The samples were pressed at 1 MPa uniaxial as a fixed load of for all bonding experiments. A heating source from HFHIF was used below a pressure of 1 × 10^−5^ torr. Different bonding temperatures of 540 °C, 570 °C, and 600 °C were achieved in one minute. A thermocouple was employed to monitor the temperature of the joint during its coupling. After cooling down the couples of Mg-Ti to ambient temperature, wire cutting tool was used to cut across the joint for each bonded sample in order to have clean bonded surfaces for corrosion analysis. Figure 2 presents (a) the optical microscopy (OM, Olympus, Tokyo, Japan) image and (b) SEM micrograph (JEOL, Tokyo, Japan) for Mg-Ti couple that was obtained at 570 °C. It is seen from the OM images that Mg surface appears darker than the surface of Ti. While, the SEM micrograph shows that Ti appears lighter than Mg and both Mg and Ti have an equal area of the surface. In order to confirm the surface morphology as well as the surface composition for the Cu and Sn inter-layer, SEM and EDX (JEOL, Tokyo, Japan) investigations were carried out. Figure 3 shows (a) SEM micrograph and (b) EDX profile analysis taken for the surface of the Mg-Ti couple that was heat treated at 570 °C. The Cu and Sn inter-layer is obvious to be homogenous with the structure of the couple. The atomic percentages for the elements found by EDX were 27.67% Mg, 71.64% Ti, 0.17% Cu, and 0.06% Sn.

### 2.4. Chemicals, Materials, and Electrochemical Cell

For electrochemical measurements, AZ31 alloy and Mg-Ti couples were welded to a copper wire then mounted in epoxy before letting to dry in air. The surface to be exposed to the test solution was ground successively with emery paper grades of increasing fineness up to 800 grit. The surface diameter of the working electrode was 1.0 cm with a total immersed surface area of 0.785 cm^2^. An electrochemical cell with three electrode configurations was used, where the samples of AZ31 and Mg-Ti coupled, a platinum foil and an Ag/AgCl were used as the working, counter and reference electrodes, respectively.

### 2.5. Weight-Loss, pH, and Electrochemical Experiments

The variations of corrosion rate (R_Corr_, mmpy), respectively with time in 5 days for the different Mg coupons in 100 cm^3^ of open to air 3.5% NaCl solutions were calculated as per the following relation [26]:(1)$$R_{\mathrm{Corr}}=\;\frac{\left(W_{1}-W_{2}\right)K}{\mathrm{DAt}}$$
where, W_1_ and W_2_ are the weighs of magnesium coupon per gram before and after its immersion in the test solution, A is the area of Mg and Mg-Ti couples per cm^2^ (A = 4.71 cm^2^), K is a constant that defines the unit of the corrosion rate (K = 8.76 × 10^4^ for the mmpy unit), D is the density per g/cm^3^ (for Mg = 1.78 g/cm^3^ and Mg-Ti = 3.122 g/cm^3^), and t is the exposure periods of time per h. The pH values were also measured along with the loss in weight using Hanna HI 4522 pH meter.

All electrochemical experiments were carried out using an Autolab Potentiostat/Galvanostat (PGSTAT20 computer controlled, Metrohm, Amsterdam, The Netherlands) operated by the general purpose software. The open-circuit potential (OCP) curves were obtained for 1.0 h as exposure time. The polarization data were collected from the cathodic side towards the anodic branch from −1900 mV to −700 mV (Ag/AgCl) at a scan rate of 1.67mV/s. The experiments of the electrochemical impedance spectroscopy (EIS) were collected at the value of OCP with a starting frequency from 100 kHz to 100 mHz at an AC wave of ±5 mV peak-to-peak overlaid. The EIS data were collected with the aid of Powersine software at 10 points per decade as the rate of change in the frequency. All electrochemical measurements have been conducted in feely aerated solutions at room temperature.

### 2.6. Surface Examinations

SEM micrographs were obtained by using a JSM-6610LV JEOL-model (Tokyo, Japan) microscope. The corroded surfaces were elementally analyzed by an energy dispersive X-ray (EDX unit installed together with the SEM machine. The SEM images and the EDX profile analyses were obtained at 15 kV and a secondary electron imaging (SEI) detector for the surface of the working electrodes.

## 3. Results and Discussion

### 3.1. OCP Curves

Figure 4 presents the potential-time curves obtained in 3.5% NaCl for (1) AZ31, (2) Mg-Ti couple treated at 540 °C, (3) Mg-Ti couple treated at 570 °C and (4) Mg-Ti couple treated at 600 °C, respectively. Curve 1 of Figure 4 indicated that the potential of AZ31 alloy is shifted in the less negative direction in the first 15 min and stays almost stable till the end of the run. Increasing potential in the positive direction was due to the formation of a surface oxide layer, while stabilizing potential values with time might have resulted from the oxide film thickening. The OCP obtained for Mg-Ti couple treated at 540 °C (curve 2) showed a slight decrease in potential at the start of measurement then gave almost similar behavior but with less negative absolute values of potential. For Mg-Ti couples treated at 570 °C and 600 °C, the potential was even shifted to lesser negative values but with the appearance of some fluctuations that increase with the increase of the heat treatment temperature of the couple. This means that coupling Ti with Mg as well as increasing the temperature of treatment increased the resistance of AZ31 alloy against uniform corrosion. Controversially, increasing the heat treatment temperature was found to increase the probability of the occurrence of pitting corrosion, which was proved by the increase of fluctuations in the potential values by the increase of treatment temperature.

### 3.2. Polarization Data

Different polarization techniques have been successfully used to report the phenomena of corrosion as well as the corrosion protection in various environments for metals and alloys [7,12,14,26]. Figure 5 shows the curves collected after 1.0 h exposure in 3.5% NaCl for (1) AZ31, (2) Mg-Ti couple treated at 540 °C, (3) Mg-Ti couple treated at 570 °C and (4) Mg-Ti couple treated at 600 °C, respectively. The data obtained from Figure 5 reveal that the recorded currents in the cathodic branch for all samples shifted towards the corrosion current density (j_Corr_) by the less negative increase of potential values. The reaction occurs on the cathode for these AZ31 alloy and Mg-Ti couples and unlike all other materials was the evolution of hydrogen even in the neutral NaCl solution as per the following equation [27,28];
2H^+^ + 2e^−^ = H_2_(2)

With increasing the potential in the positive direction in the anodic side, the current increased because of the dissolution of Mg releasing the electrons that would be consumed at the cathode. The increase of currents is due to the occurrence of corrosion of Mg as per the following reactions [29];
2Mg = 2Mg^2+^ + 4e^−^(3)

The magnesium cations (Mg^2+^) are very reactive and would form Mg(OH)_2_ after reacting with OH^−^ group from the test electrolyte. MgO formation takes place during the potential scan in the anodic branch as per the following equation [29];
2Mg^2+^ + 2OH^−^ = Mg(OH)_2_(4)
Mg + O = MgO(5)

The formation of Mg(OH)_2_ and even MgO does not provide any protection to the surface from being attacked by NaCl solution and corroded. That is why the obtained current for all tested materials are increasing with scanning the potential in the anodic side.

Table 1 lists the parameters obtained from the measurements shown in Figure 5 and Figure 6. These parameters are βc, βa (cathodic and anodic Tafel slopes), E_Corr_ (corrosion potential), j_Corr_, corrosion rate (R_Corr_), and R_P_ (polarization resistance). The values recorded in Table 1 were obtained as reported in the previous research [25,28,30]. The curves of Figure 5 and the data listed in Table 1 indicated that the coupling of Ti with Mg at 540 °C greatly reduced j_Corr_ and R_Corr_ values and raised R_P_ values. Increasing the heat treatment of the Mg-Ti couple from 540 °C to 570 °C and further to 600 °C was remarkably found to decrease the values of j_Corr_, R_Corr_ and increased the value of R_P_.

Figure 6 shows the potentiodynamic polarization curves recorded after prolonging the time of exposure to 48 h in 3.5% NaCl for (1) AZ31, (2) Mg-Ti couple treated at 540 °C, (3) Mg-Ti couple treated at 570 °C and (4) Mg-Ti couple treated at 600 °C, respectively. The obtained curves showed less current values compared to those currents collected for the polarization curves of the different samples after 1.0 h immersion. Treating the Mg-Ti couple at 540 °C further decreased the corrosion parameters seen in Table 1 because of raising the corrosion resistance of the couple. Increasing the treatment temperature to 570 °C further raised the corrosion resistance, which was greatly increased when the temperature of treatment was increased to 600 °C. This was also confirmed by the lower cathodic and anodic currents, lower values of j_Corr_ and R_Corr_ and higher values of R_P_. Prolonging the time to 48 h before measurement thus increases the corrosion resistance of the Mg alloy as well as the different Mg-Ti couples.

### 3.3. EIS Measurements

The EIS spectra were collected after 1 h and 48 h immersion in the sodium chloride solutions prior to measurement. Figure 7 shows the Nyquist spectra recorded for (1) AZ31, (2) Mg-Ti couple treated at 540 °C, (3) Mg-Ti couple treated at 570 °C, and (4) Mg-Ti couple treated at 600 °C, respectively after 1 h immersion in test solutions. The Nyquist spectra were also collected after 48 h as shown in Figure 8. All these impedance spectra were analyzed via fitting obtained data to the circuit model presented in Figure 9. The parameters calculated from the circuit of Figure 9 are depicted in Table 2. It is known that R_S_ is the resistance of the solution. Q_1_ and Q_2_ are the first and second constant phase elements (CPEs). Also, R_P1_ and R_P2_ are the first and second polarization resistances; here R_P1_ refers to a resistance of the film or the layer formed on the surface of AZ31 and Mg-Ti couples, while R_P2_ represents the polarization resistance at the surface of our working electrode alloy samples.

It is indicated from Figure 7; Figure 8 that there is only once semicircle obtained for all tested samples; the diameter obtained for AZ31 was the smallest. The diameter of the semicircle is clear to increase for the Mg-Ti compared to AZ31; further increases are noted with the increase of the heat treatment temperature. The parameters presented in Table 2 also confirmed that the values of all resistances, R_S_, R_P1_, and R_P2_ were the lowest for AZ31; these resistances increased for the couples and up on the increase of its heat treatment temperature. It is known that the values of R_P1_ and R_P2_ here if increased can be considered as a decrease in the uniform corrosion rate, which is opposed to tendency towards pitting attack. Moreover, the increase of these resistance values proves that the corrosion of the couples are lower than that for AZ31 and the increase of the heat treatment temperature provided more decreases in the corrosion. The values recorded for Q_1_ and Q_2_ (CPEs) and their n component values circa unity (see Table 2) can be considered as double layer capacitors (C_dl_), which have some pores.

Extending the exposure time to 48 h before measurement (Figure 8) decreased the corrosion of the tested materials. This was confirmed by the wider diameter for their semicircles as compared to those collected after only 1 h immersion (Figure 7). Moreover, the values of R_S_, R_P1_, and R_P2_ are higher, while the values of Q_1_ and Q_2_ are also lower than those obtained for AZ31 alloy and this effect further increase with increasing the temperature of the heat treatment. The results obtained from EIS results thus in good agreement with the data obtained by polarization, where the resistance against corrosion for the different Mg-Ti couples increased with increasing the heat treatment temperature as well as increasing the immersion time from 1 h to 48 h.

### 3.4. Weight-Loss, pH, SEM Images, and EDX Investigations

Figure 10 illustrates the variations (a) weight-loss (∆W) and (b) corrosion rate (R_Corr_) as a function of time for (1) Mg AZ31 alloy, (2) Mg-Ti couple treated at 540 °C, (3) Mg-Ti couple treated at 570 °C, and (4) Mg-Ti couple treated at 600 °C, in 3.5% NaCl solutions. One can see easily that the value of ∆W increases and the values of R_Corr_ decreases with the increase of time for all coupons. The increase of ∆W is due to the increase of dissolution of Mg and Mg-Ti couples with increasing the time of immersion. The increase of ∆W for AZ31 was higher than those recorded for Mg-Ti couples, which its loss in weight decreases with the increase of heat treatment temperature. On the other hand, the values of R_Corr_ decrease due to the accumulation of corrosion products on the surface with increasing the time of immersion. It is also seen that the highest R_Corr_ values were obtained for AZ31 alloy (curve 1) followed by the couples that were treated at lower temperatures, i.e., 540 °C > 570 °C > 600 °C. The increase of weight-loss with time is due to the dissolution of the surface with time, while the decrease of corrosion rate comes from the accumulation of the corrosion products on the surface of the material leading to reducing the corrosion. This is in good agreement with the principles of the occurrence of uniform corrosion, which starts fast and decrease with time.

On the other hand, the change of pH values, which were obtained for the solutions during performing the weight-loss experiments for the different materials over 5 days immersion are shown in Figure 11. It is clear that the pH values initially increased from 6.22 (pH of the chloride solution without having materials immersed in) after 24 h of the immersion of the different materials for all samples and it was the highest for AZ31 alloy and decreased for the Mg-Ti couples, particularly with the increase of heat treatment temperature. The values of pH for all solutions were noticed to decrease with increasing immersion time, which is most probably due to the decrease of corrosion rate for our materials with time. This is also because the corrosion process is accompanied by a surface alkalinization that could lead to the initial increase of pH with time as reported in the earlier studies [26,31]. Therefore, the increase of corrosion rate due to the fast dissolution of materials after short immersion periods of time would lead to the increase of pH values, which would then decrease when the corrosion rate slows down due to the coverage of surface of with thick corrosion product layer. Furthermore, the increase of the heat treatment temperature led to further reduction in pH values as compared to the blank alloy (AZ31) that suffers more corrosion that could lead to raising the values of pH as well as the values R_Corr_ Also, the increase of heat treatment temperature from 540 °C to 600 °C decreased the weight loss with time and therefore the values of R_Corr_. These weight-loss, R_Corr_ and pH measurements are in good agreement with the electrochemical results that the corrosion of these materials decreased as per the following order; AZ31 > Mg-Ti couple treated at 540 °C > Mg-Ti couple treated at 570 °C > Mg-Ti couple treated at 600 °C.

The SEM micrographs and the EDX analyses were carried out to see the morphology as well as the surface analysis for the different tested samples after its immersion for longer period of time in the sodium chloride. Figure 12 shows (a) SEM images and (b) EDX spectra for the AZ31 alloy after 5 days immersion in 3.5% NaCl solution. Immersing the Mg AZ31 alloy in the solution of 3.5% NaCl for longer time led to the occurrence of both uniform corrosion and pitting attack. At this condition, the severity of the uniform corrosion is reduced as a result of the formation of corrosion products on the surface of the alloy by its immersion in the NaCl solution. The pitting corrosion also took place through the development of pits at the places that the aggressive Cl^−^ that present in the solution displace the adsorbed oxygen from the surface of the alloy. The displacement of O^−2^ by Cl^−^ takes place due to the small diameter of Cl^−^ that would permit the penetration of these ions through the film formed on the surface displacing O^−2^ at the weakest metal-oxygen bond [32]. The elements found on the surface of AZ31 alloy with its weight percentages recorded 36% Mg, 51.13% O, 5.62% Na, and 6.90% Cl. This proves that the compound formed on the surface of the alloy is MgO in addition to the deposited NaCl salt.

Figure 13 depicts the SEM image and EDX spectrum for Mg-Ti couple treated at 540 °C, after its immersion for 5 days in 3.5% NaCl solution. It is obvious from the SEM image that the part belongs to Mg is deteriorated, while the part for Ti keeps protected and this is because the occurrence of galvanic corrosion between Mg and Ti of the couple. Here, the severity of galvanic corrosion is the maximum because the potential difference between Mg and Ti is very high. The standard potentials of Mg and Ti can be expressed according to their reduction reactions as following [33,34,35];
Mg^2+^ + 2e^−^ ↔ Mg(s) (E⁰ = −2.356 V)
Mg(OH)_2_(s) + 2e^−^ ↔ Mg(s) + 2OH^−^ (E⁰ = −2.687 V)
Ti^2+^ + 2e^−^ ↔ Ti(s) (E⁰ = −0.163 V)
Ti^3+^ + e^−^ ↔ Ti^2+^ (E⁰ = −0.37 V)

This difference in the potential between Mg and Ti led to increasing the corrosion of Mg and protects Ti from being corroded. The EDX profile analyses taken for Mg area shown in Figure 13 recorded 33.92% Mg, 48.88% O, 13.66% Cl, 2.61% Na, and 0.93% Al. This means the compounds that were formed on the surface of Mg after its corrosion could be MgO, MgCl_2_, and NaCl salt. On the other hand, EDX profile taken for most of the surface of Ti area (Figure 13) provided 18.64% Mg, 48.38% O, 25.17% Ti, 3.78% Na, 1.20 Cl, 1.59% Al, and 1.24% V. The low percentage of Ti proves the coverage of the surface of Ti with a corrosion product layer. This layer composed mainly from MgO because the percentages of both Mg and O are high.

In order to confirm that the surface of Ti is more protected due to its coupling to Mg and heat treated at 540 °C, SEM/EDX analysis were taken for the surface of Ti as shown in Figure 14. It is seen that the surface shows two areas; one is bare and the other is has thick corrosion products. The EDX spectrum obtained from the bare surface recorded 73.24% Ti, 15.73% O, 4.75Al, 3.24% V, 0.67% Na, and 0.27 Cl. It can be understood that this area of the surface consists mainly of pure Ti, an oxide and some NaCl salt deposited from the solution on the surface. On the other side, the EDX of the surface area that has thick layer on it recorded 43.70% O, 34.23% Mg, 10.94% Cl, 8.26% Na, 0.21% V, and only 2.66% Ti. The low percentage of Ti along with the high percentages of O and Mg indicate that the surface contains mainly Mg oxide. Also, the presence of high amounts of sodium and chlorine proves the presence of sodium chloride salt on the surface.

The SEM image and EDX spectrum taken for the surface of the Mg-Ti couple that was treated at 570 °C is presented in Figure 15. Due to the occurrence of galvanic corrosion, Mg dissolves rapidly and Ti stays not being corroded. EDX profile analyses proved that Mg part suffers severe corrosion as the wt% of Mg was 33.48% and the rest was O, Na, and Cl, in addition to a trace of Al. While, the EXD profile taken for the part of Ti gives indication on the presence of pure Ti with some products of corrosion were formed on the surface of Ti. This was further proved by the SEM and EDX analyses shown in Figure 16, which were taken for Ti part of the Mg-Ti couple. It is obvious thus a thick layer of products partially covers the surface with the presence of some places that are bare from being covered. The wt% obtained for the corrosion products area were as following; 20.90% Mg, 27.33% O, 32.92% Cl, 18.52% Na, and only 0.32% Ti. The very low wt% of Ti and the high wt% of Mg and O as well as the Cl and Na indicate that the corrosion products are mainly MgO and thick layer of NaCl salt. The EDX profile analysis taken for the bare area reveals that this area of the surface belongs to the pure Ti and TiO_2_ because its wt% recorded 62.52%. Other elements found in the bare area with their wt% were 23.18% O, 4.65% Mg, 3.68% Al, 2.05% Na, and 1.20% Cl.

The SEM/EDX investigations taken for the surface of the couple of Mg-Ti that was treated at 600 °C are shown in Figure 17. It is seen that the Mg of the Mg-Ti couple suffers severe dissolution in the NaCl solution as a result of galvanic corrosion that gets accelerated as a result of the coupling to a noble metal such as Ti. The deteriorated surface of Mg that appears in the SEM micrograph (Figure 17) has shown 36.37% Mg, 48.77% O, 8.20% Na, and 6.66% Cl. This indicates that the surface is fully covered by the MgO and NaCl salt; here the formed MgO does not provide any protection to the surface of Mg as well as the Mg-Ti couple. The elements found on the surface of Ti appears in Figure 17 were as following; 25.20% Mg, 42.15% O, 14.86% C, 8.01% Na, 9.47% Cl, and only 0.18% Ti. The surface of Ti thus is fully covered with thick layer of MgO and NaCl salt. This confirms that the couple corrodes through only the dissolution of Mg and the protection of Ti.

The SEM/EDX of the surface for Ti part of the Mg-Ti couple and that was treated at 600 °C and immersed for 5 days are presented by Figure 18. The SEM micrograph shown in Figure 18 has two parts; one is small and has no any layers to cover it, while the second part is large and contains mainly of corrosion products. The wt% of the elements found on the surface of the bare part of Ti as shown in Figure 18, were as following; 62.04% Ti, 20.76% O, 4.75% Mg, 3.48% Na, 2.46% Cl, and traces of Al and V. The surface here must be pure Ti with some MgO as a corrosion product and also NaCl salt that was deposited from the solution. On the other hand, the elements found on the surface of the large area as indicated by EDX profile analysis taken were, 13% Mg, 28.25% O, 12.03% Ti, 7.00% Na, 8.37% Cl and traces of Al and V. A great part of the surface thus has massive corrosion product layer that contain MgO and NaCl deposited salt.

The SEM/EDX investigations thus confirm the data obtained by the electrochemical techniques and the mechanism of the corrosion process can be represented by the schematic diagram shown in Figure 19. Here, the dissolution of Mg takes place upon the increase of potential in the anodic direction as indicated from the polarization measurements and expressed by Equation (3) releasing Mg^2+^. These cations react rapidly with hydroxyl groups presented in the solution to from Mg(OH)_2_, which is not protective enough to prevent the corrosion of Mg alloy and Mg-Ti couples in the chloride solution as depicted by Equation (4). The formation of MgO also occurs on the surface of the test electrode as a result of the reaction between Mg and O (see Equation (5)). However, the formation of this oxide does not protect the surface from being corroded under the aggressiveness action of the concentrated chloride solution.

## 4. Conclusions

This work reported the manufacturing of Mg-Ti couple at 540 °C, 570 °C, and 600 °C heat treatment temperatures. The corrosion of the heat treated Mg-Ti couples was carried out as well as compared to the corrosion of AZ31 Mg alloy in NaCl solutions. The corrosion resistance of AZ31 alloy was found to be the lowest. This corrosion resistance was greatly increased when coupling the Mg with Ti and was further increased with the increase of the temperature of the heat treatment from 540 °C to 570 °C and further to 600 °C. Prolonging the time of immersion from 1.0 to 48 h was also found to increase the corrosion resistance for all tested samples as was indicated by the OCP and polarization measurements. SEM micrographs and EDX analyses confirmed that Mg-Ti couple suffers severe corrosion, raising the heat treatment and increasing the immersion time remarkably decreases this corrosion. All results were in good agreement and confirmed that the dissolution of Mg-Ti takes place via its galvanic corrosion due to the deterioration of Mg and the protection of Ti.

## Figures and Tables

**Figure 1 materials-12-01300-f001:**
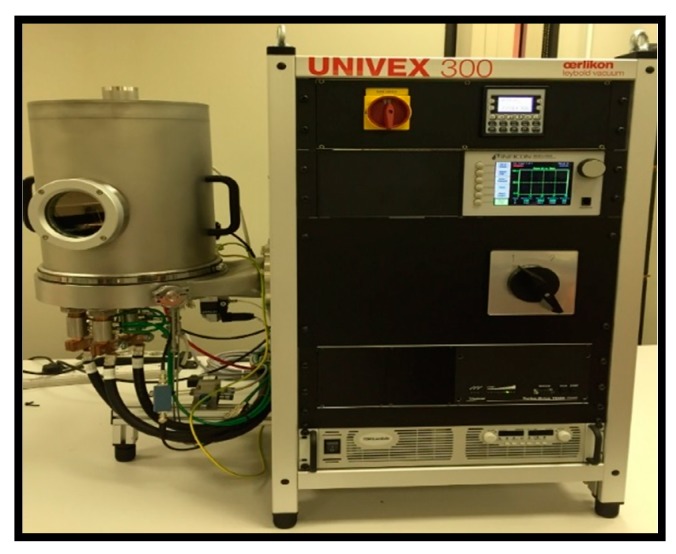
Lecybold coating system UNIVEX 300.

**Figure 2 materials-12-01300-f002:**
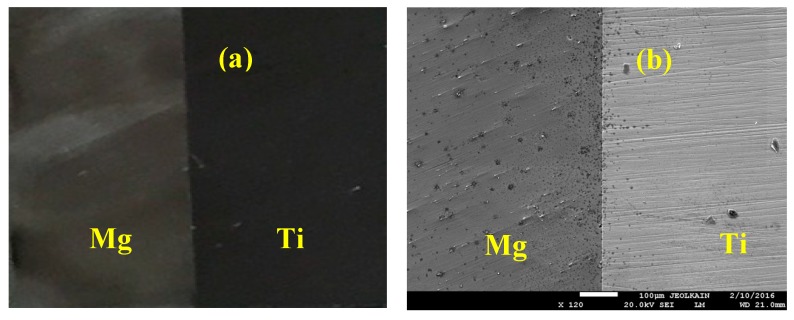
(**a**) Optical microscopy (OM) and (**b**) SEM images show a sample of the fabricated Mg-Ti couple.

**Figure 3 materials-12-01300-f003:**
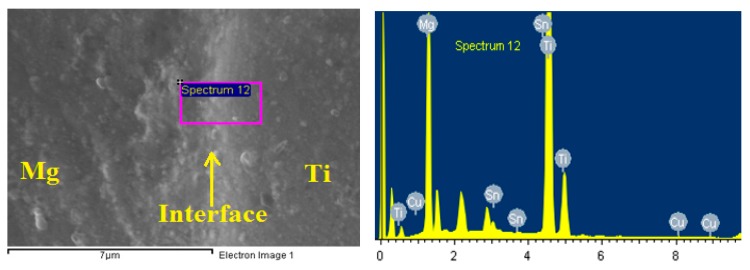
SEM micrograph and EDX spectrum for Mg-Ti couple that was heat treated at 570 °C.

**Figure 4 materials-12-01300-f004:**
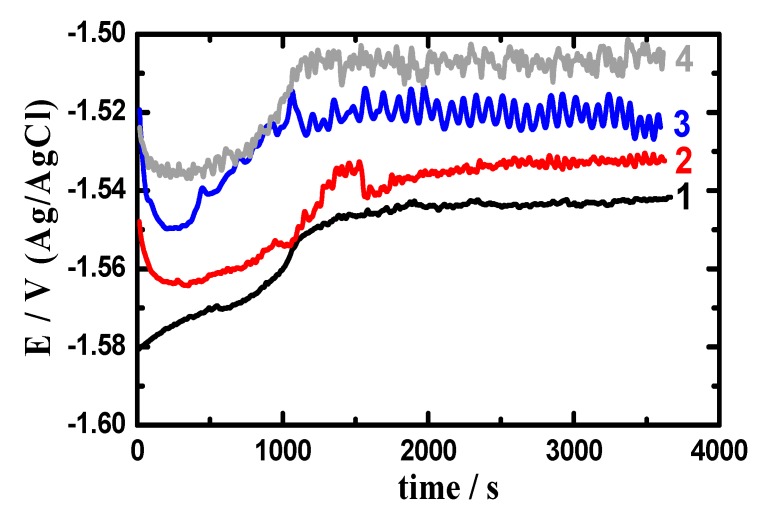
Open-circuit potential (OCP) curves obtained in 3.5% NaCl for (1) AZ31, (2) Mg-Ti couple treated at 540 °C, (3) Mg-Ti couple treated at 570 °C, and (4) Mg-Ti couple treated at 600 °C, respectively.

**Figure 5 materials-12-01300-f005:**
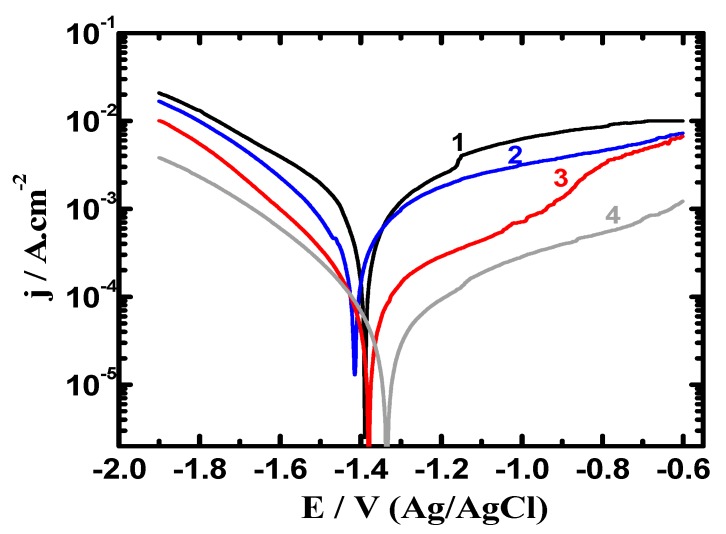
Potentiodynamic polarization collected for (1) AZ31, (2) Mg-Ti couple treated at 540 °C, (3) Mg-Ti couple treated at 570 °C and (4) Mg-Ti couple treated at 600 °C, respectively after their immersion in 3.5% NaCl for 1.0 h.

**Figure 6 materials-12-01300-f006:**
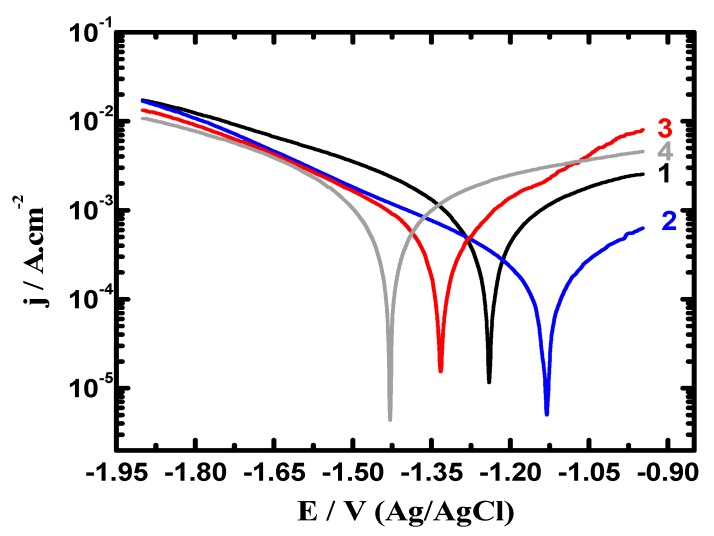
Polarization curves collected after 48 h exposure in 3.5% NaCl for (1) AZ31, (2) Mg-Ti couple treated at 540 °C, (3) Mg-Ti couple treated at 570 °C, and (4) Mg-Ti couple treated at 600 °C, respectively.

**Figure 7 materials-12-01300-f007:**
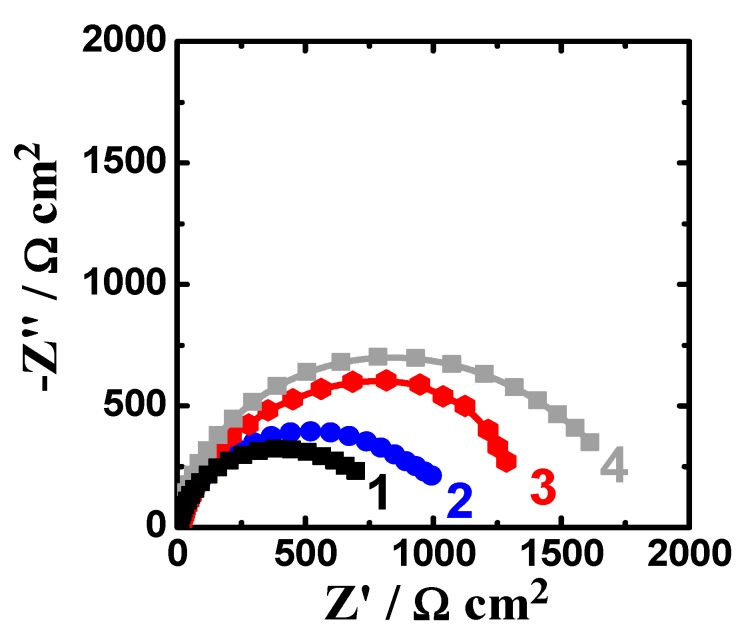
Nyquist spectra recorded for (1) AZ31, (2) Mg-Ti couple treated at 540 °C, (3) Mg-Ti couple treated at 570 °C and (4) Mg-Ti couple treated at 600 °C, respectively after 1.0 h immersion in 3.5% NaCl.

**Figure 8 materials-12-01300-f008:**
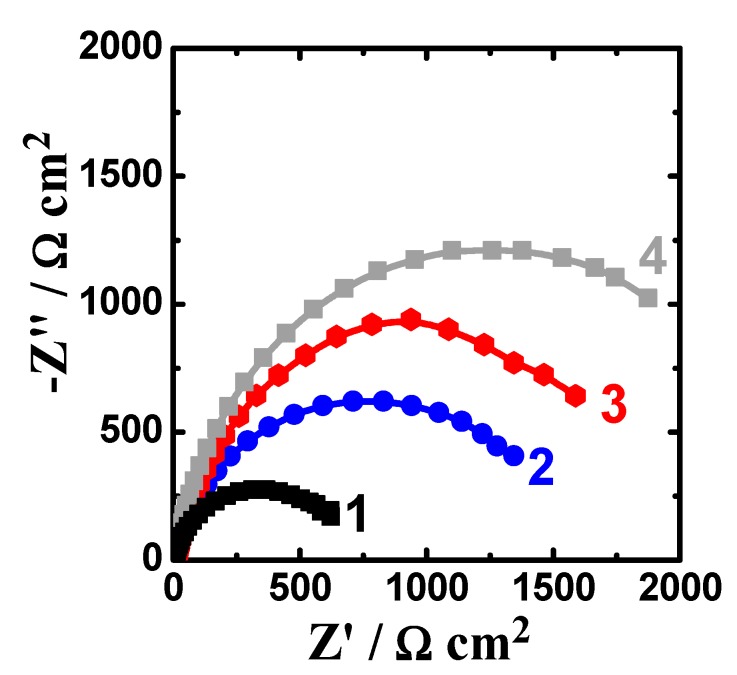
Nyquist spectra collected after 48 h exposure to 3.5% NaCl for (1) AZ31, (2) Mg-Ti couple treated at 540 °C, (3) Mg-Ti couple treated at 570 °C, and (4) Mg-Ti couple treated at 600 °C, respectively.

**Figure 9 materials-12-01300-f009:**
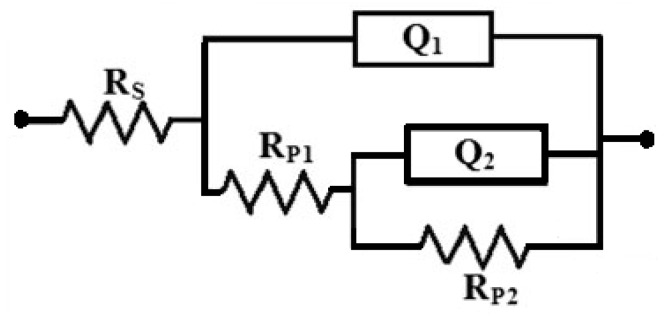
Employed fitting circuit for the electrochemical impedance spectroscopy (EIS) spectra shown in Figure 7 and Figure 8.

**Figure 10 materials-12-01300-f010:**
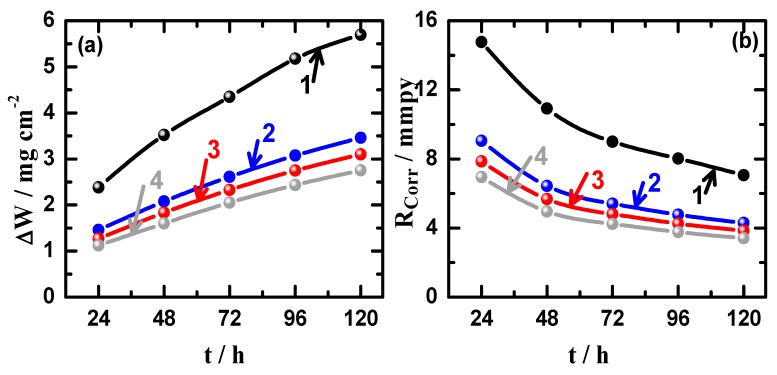
Variations of (**a**) weight-loss (∆W) and (**b**) corrosion rate (R_Corr_) as a function of time for (1) Mg AZ31 alloy, (2) Mg-Ti couple treated at 540 °C, (3) Mg-Ti couple treated at 570 °C, and (4) Mg-Ti couple treated at 600 °C, in 3.5% NaCl solutions.

**Figure 11 materials-12-01300-f011:**
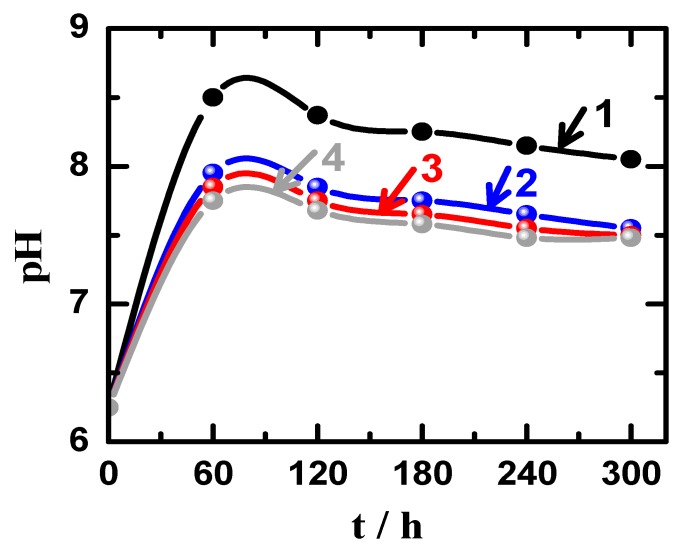
Variations of pH with time for (1) Mg AZ31 alloy, (2) Mg-Ti couple treated at 540 °C, (3) Mg-Ti couple treated at 570 °C, and (4) Mg-Ti couple treated at 600 °C, in 3.5% NaCl solutions.

**Figure 12 materials-12-01300-f012:**
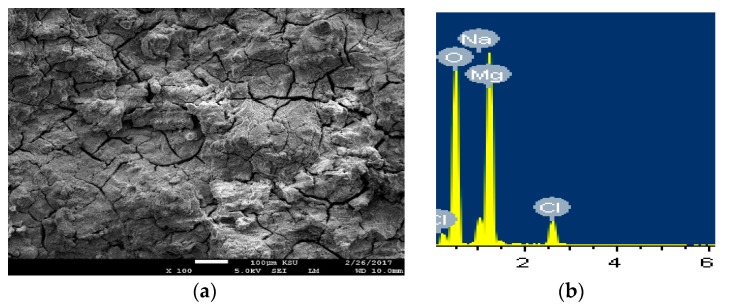
(**a**) SEM image and (**b**) EDX spectrum for the alloy AZ31 after 5 days in 3.5% NaCl solution.

**Figure 13 materials-12-01300-f013:**
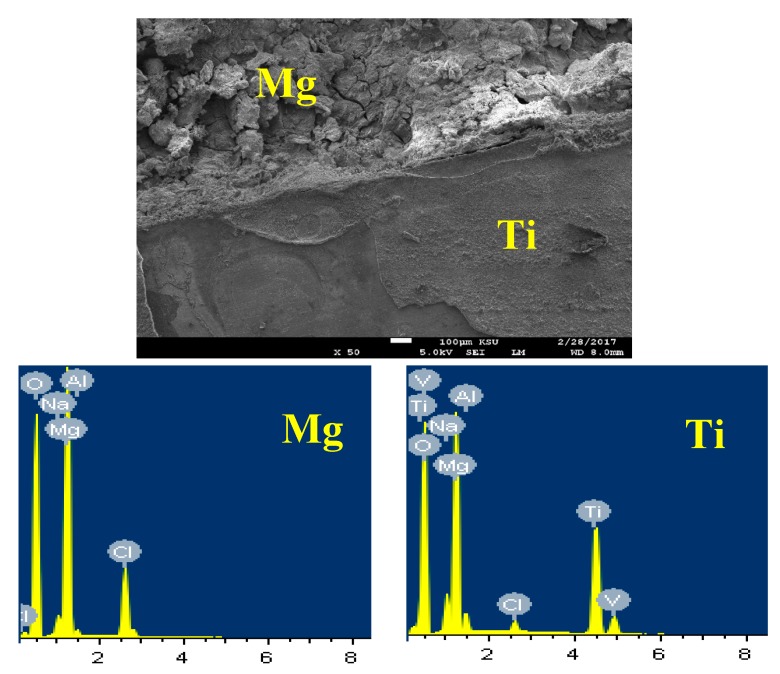
SEM image and EDX spectrum collected for the Mg-Ti couple that was treated at 540 °C and immersed for 5 days in NaCl solution.

**Figure 14 materials-12-01300-f014:**
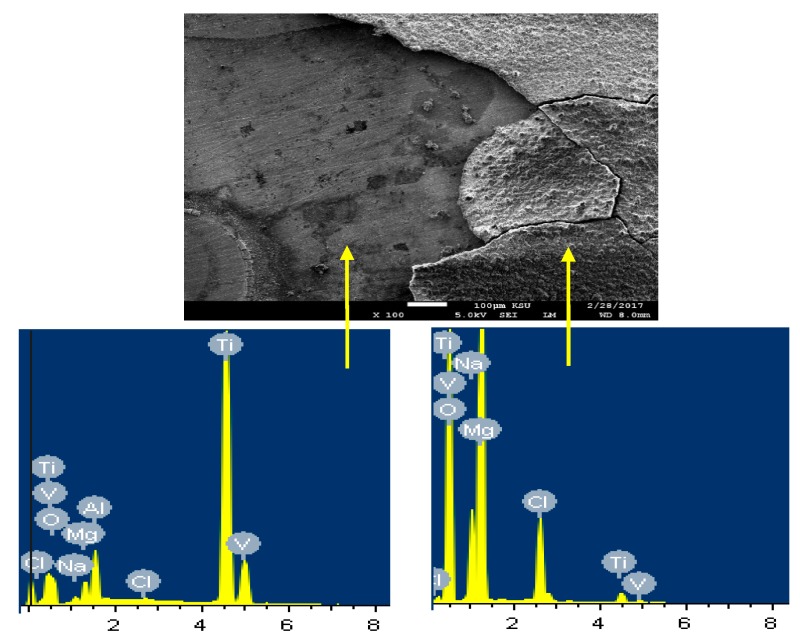
SEM image and EDX spectrum for the Ti part of the Mg-Ti couple that was treated at 540 °C (immersion time = 5 days).

**Figure 15 materials-12-01300-f015:**
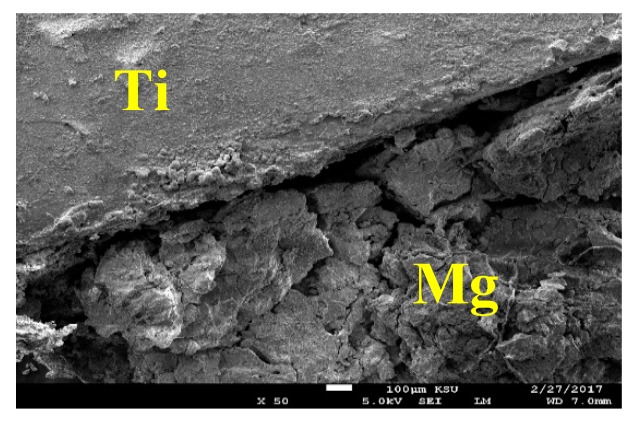
SEM image and EDX spectrum for Mg-Ti couple treated at 570 °C (immersion time = 5 days).

**Figure 16 materials-12-01300-f016:**
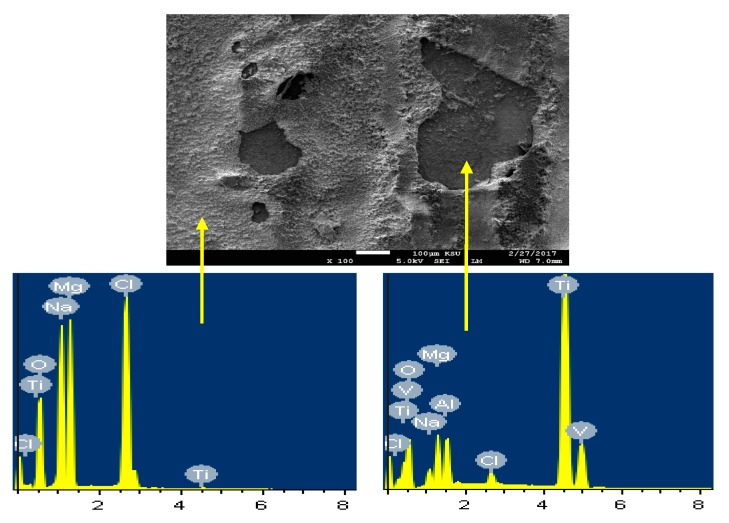
The SEM micrograph and EDX profile analyses for the Ti part of the Mg-Ti couple that was treated at 570 °C (immersion time = 5 days).

**Figure 17 materials-12-01300-f017:**
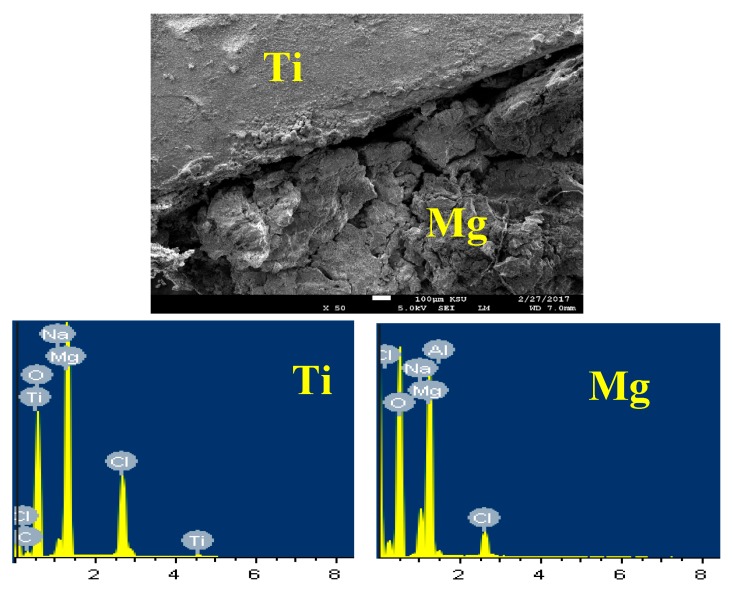
SEM image and EDX spectrum for Mg-Ti couple treated at 600 °C (immersion time = 5 days).

**Figure 18 materials-12-01300-f018:**
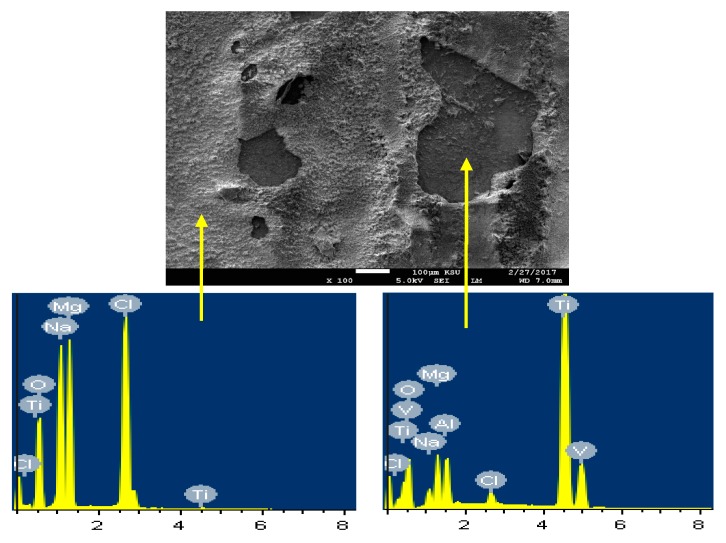
SEM image and EDX spectrum for the Ti part of the Mg-Ti couple that was treated at 600 °C (immersion time = 5 days).

**Figure 19 materials-12-01300-f019:**
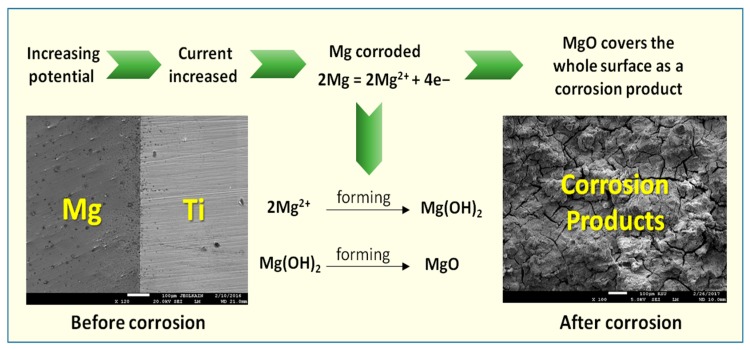
Schematic diagram for the corrosion mechanism at the surface of Mg-Ti couples after being immersed in the chloride solution.

**Table 1 materials-12-01300-t001:** Parameters collected for the different Mg-Ti couples after various immersion periods of time in the chloride test solutions.

Sample	β_c_/mVdec^−1^	E_Corr_/mV	β_a_/mVdec^−1^	j_Corr_/µAcm^−2^	R_p/_Ωcm^2^	R_Corr_/mmpy
AZ31 (1 h)	160	−1373	165	675	0.078	9.47
Mg-Ti 540 °C (1 h)	155	−1400	160	360	0.129	5.05
Mg-Ti 570 °C (1 h)	165	−1365	155	75	0.572	1.05
Mg-Ti 600 °C (1 h)	170	−1330	140	35	1.115	0.49
AZ31 (48 h)	165	−1382	130	480	0.076	6.73
Mg-Ti 540 °C (48 h)	155	−1420	140	260	0.132	3.65
Mg-Ti 570 °C (48 h)	150	−1370	135	62	0.535	0.87
Mg-Ti 600 °C (48 h)	145	−1335	130	23	1.371	0.32

**Table 2 materials-12-01300-t002:** EIS parameters obtained from EIS measured data.

	Parameter						
Solution	R_S_/Ω cm^2^	Q_1_		R_P1_/Ω cm^2^	Q_2_		R_P2_/Ω cm^2^
		Y_Q1_/µF cm^−^^2^	n		Y_Q2_/µF cm ^−^^2^	n	
AZ31 (1 h)	9.99	0.8198	1.00	356	0.2223	0.91	1423
Mg-Ti 540 °C (1 h)	12.43	0.5632	1.00	469	0.1996	0.89	1789
Mg-Ti 570 °C (1 h)	13.25	0.2914	1.00	655	0.1182	0.87	2964
Mg-Ti 600 °C (1 h)	15.12	0.05196	1.00	833	0.06801	0.94	3198
AZ31 (48 h)	12.49	0.1556	0.89	538	0.5321	0.94	669
Mg-Ti 540 °C (48 h)	21.26	0.083	0.90	732	0.4753	0.81	2125
Mg-Ti 570 °C (48 h)	16.65	0.2604	0.80	891	0.0924	0.84	2720
Mg-Ti 600 °C (48 h)	19.78	0.021	0.92	927	0.0258	087	2963

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
