# Peer review of "Manufacturing of Mg-Ti Couples at Different Heat Treatment Temperatures and Their Corrosion Behavior in Chloride Solutions"

_materials, 2019, doi:10.3390/ma12081300_

Reviewer 1 Report

Through the manufacturing of  Mg-Ti couples at different heat treatment temperatures, the material couples and corrosion behaviors were studied. The following problems should be fixed before pubulication:

Replace the "new" in the title and related content.

More background and process on the related area of metallic couples should be added in the introduction;

The Cu and Sn interlayer should be characterized with morphology and compositions;

The test parameters should be detailed in the electrochemical test methods, e.g. the surface area and tested location should be indicated on the coupled material.

During the corrosion, the pH values, mass loss, and hydrogen release should be monitored to caculate the corrosion rates.

A schematic diagram is strongly suggested to discuss the corrosion behavior of the material, particularly at the interface.

Author Response

Response to Reviewer#1

Reviewer’s General Comments: Moderate English changes required and other items that can be improved and must be improved.

Author Response to General Comments: Dear Reviewer, all authors do really appreciate your all comments, suggestions, and recommendations. We believe that it all improves our manuscript, so thank you very much for your efforts and valuable advices. We have carefully revised our manuscript as per your valuable suggestions and recommendations and the revised areas are highlighted in yellow in the manuscript. In response to your comments, we reproduced all your comments by heading “Reviewer Comment” and our responses are given by heading “Response to Reviewer”.

Reviewer Comment:  Replace the "new" in the title and related content.

Response to Reviewer: As recommended the word “new” has been omitted from the Title and there was no “new” in the whole text of the manuscript to be replaced or omitted.

Reviewer Comment: More background and process on the related area of metallic couples should be added in the introduction;

Response to Reviewer: As recommended, more background and process on the related area of metallic couples have been added and highlighted in the Introduction part.

Reviewer Comment:  The Cu and Sn interlayer should be characterized with morphology and compositions;

Response to Reviewer: We have extensively investigated the surface morphology and the chemical analysis of all tested samples. It was confirmed that the formation of a thick layer of the corrosion products led to a complete coverage of the surface with which the detection of Cu and Sn was impossible. The SEM/EDX part is 7 figures, 6 figures were on the couples and were taken on different areas. None of these areas showed any presence for either Sn or Cu because of the formed corrosion products are thick and are mainly Mg compounds.

Reviewer Comment: The test parameters should be detailed in the electrochemical test methods, e.g. the surface area and tested location should be indicated on the coupled material.

Response to Reviewer: Done, as has been highlighted in the experimental section “2.4. Chemicals, materials and electrochemical cell”.

Reviewer Comment: During the corrosion, the pH values, mass loss, and hydrogen release should be monitored to calculate the corrosion rates.

Response to Reviewer: Dear Reviewer, this is very critical comment. As for measuring the values of pH, truly it did not come to our mind to measure the pH values, particularly after performing the measurements after 48 h of the electrodes immersion. However, it is expected that the pH values will decrease from the near neutral to the acidic values due to the dissolution of Mg, which leads to the release of hydrogen as a cathodic reaction. As for the mass loss and as stated in our experimental part “subsection: 2.4. Chemicals, materials and electrochemical cell”, the exposed area of the working electrode was relatively small with which the loss in weight would be very small. As for measuring the hydrogen release, we have already stated in the experimental part “subsection: 2.5. Electrochemical measurements” all measurements were carried out in freely aerated (open to air) NaCl solutions and also because of the small exposed surface area of the working electrode, the released hydrogen would be negligible. We hope that this answer explains the condition for not being able to include the mentioned measurements to the manuscript and is enough to be accepted by you.

Reviewer Comment: A schematic diagram is strongly suggested to discuss the corrosion behavior of the material, particularly at the interface.

Response to Reviewer: We have implanted a schematic diagram (as depicted below) for the corrosion mechanism at the surface of the coupled alloys after being immersed in the chloride solution. For that we have added “The SEM/EDX investigations thus confirm the data obtained by the electrochemical techniques and the mechanism of the corrosion process can be represented by the schematic diagram shown in Figure 16.” to the text in order to explain the reason for adding one more Figure. For your kind information, the corrosion products cover the whole surface and prevent the discovery of Cu and Sn. The presence of Cu and Sn between the two couples was to facilitate the diffusion.

Figure 16. Schematic diagram for the corrosion mechanism at the surface of the coupled alloys after being immersed in the chloride solution. (can be seen in the manuscript)

Reviewer 2 Report

You have done several electrochemical investigations on  three Mg-Ti couples  and compared with AZ31 alloy  after 1.0 h and 48 h immersion in 3.5% NaCl solutions.

Please  describe your measurements in more detail.

For which temperature electrochemical tests were conducted

Did you perform your measurements in an oxygen saturated electrolyte when you started and did you aerate the electrolyte during the measurements (as dissolved oxygen in the electrolyte plays an important on corrrosion)?

 Where did you start your potentiodynamic polarization measurements - at Ecorr and then go 250 mV in negative direction and then 500 mV in positive direction (to come to Ecorr + 250 mV) or did you start in anodic direction?

Did you do the linear polarization measurements with the same samples after you did OCP measurements or did you use new samples for each test?

Author Response

Response to Reviewer#2

Reviewer’s General Comments: English language and style are fine/minor spell check required & You have done several electrochemical investigations on three Mg-Ti couples and compared with AZ31 alloy after 1.0 h and 48 h immersion in 3.5% NaCl solutions.

Author Response to General Comments: Dear Reviewer, thank you very much for reviewing, giving constructive comments, and guiding us to highly improve our present work. We would like to let you know that we have carefully revised our manuscript as per your valuable suggestions and recommendations and the revised areas are highlighted in yellow. In response to your comments, we reproduced all your comments by heading “Reviewer Comment” and our responses are given by heading “Response to Reviewer”.

Reviewer Comment:  Please describe your measurements in more detail.

Response to Reviewer: Our measurements have been described in more details as suggested, please see the highlighted areas in the experimental part of the manuscript.

Reviewer Comment:  For which temperature electrochemical tests were conducted.

Response to Reviewer: All electrochemical measurements have been conducted in freely aerated solutions at room temperature. We have already added this as the last sentence to the experimental part “section 2.5. Electrochemical measurements”.

Reviewer Comment:  Did you perform your measurements in an oxygen saturated electrolyte when you started and did you aerate the electrolyte during the measurements (as dissolved oxygen in the electrolyte plays an important on corrosion)?

Response to Reviewer: Our experiments were carried out in an open to air solutions without aeration before or during the measurements. We understand exactly your point as we have got long experience with studying the corrosion in aerated, de-aerated and freely aerated solutions. For that and as per your inquiry, we added the last sentence to the experimental part “section 2.5. Electrochemical measurements”.

Reviewer Comment: Where did you start your potentiodynamic polarization measurements - at Ecorr and then go 250 mV in negative direction and then 500 mV in positive direction (to come to Ecorr + 250 mV) or did you start in anodic direction?

Response to Reviewer: Our polarization measurements started from the anodic direction, exactly from -1900 mV towards the less negative direction to -700 mV. We stated that and highlighted it in the experimental part “subsection: 2.5. Electrochemical measurements”. As “The potentiodynamic polarization data were obtained by scanning the potential in the forward direction from the cathodic side towards the anodic branch from -1900 mV to -700 mV versus Ag/AgCl at a scan rate of 1.67mV/s.”

Reviewer Comment: Did you do the linear polarization measurements with the same samples after you did OCP measurements or did you use new samples for each test?

Response to Reviewer: As we stated before we did not use Linear Polarization technique but our measurements were what is called “Forward Anodic Polarization” in which the measurements start with scanning the potential from the cathodic potential (more negative or less positive value) directly towards the anodic direction (the less negative or the more positive values). For all measurements including the potentiodynamic polarization one, new surface (new polished (ground) sample) as well as a fresh portion of the test solution were used.

Round  2

Reviewer 1 Report

Some comments were replied with good revisions, but the major problems have not been addressed after the revisions. I suggest another major revision:

 The Cu and Sn inter-layer before the corrosion test should be characterized with morphology and compositions;

The corrosion rate could be directly compared with the pH value changes, mass loss, and hydrogen release, which are still strongly suggested, particularly when the authors think " it is expected that the pH values will decrease from the near neutral to the acidic values due to the dissolution of Mg".

A more detailed corrosion mechanism should be included in the schematic diagram (Fig. 16).

Author Response

Response to Reviewer#1

Dear Reviewer, thank you very much for efforts in reviewing our manuscript. We have performed all experiments you suggested and very carefully replied to the comments you raised one by one, improved English language, improved the Methodology, Results and discussion (the whole manuscript is almost restructured), in the hope that our work gets your positive recommendation.

Reviewer Comment: The Cu and Sn inter-layer before the corrosion test should be characterized with morphology and compositions;

Response to Reviewer: Done as has been highlighted in page 3 and page 4, line 108 to line 115. A new figure was inserted (Figure 3) and more text was implanted into the Results and discussion part as follows,

In order to confirm the surface morphology as well as the surface composition for the Cu and Sn inter-layer, SEM and EDX investigations were carried out. Figure 3 shows (a) SEM micrograph and (b) EDX profile analysis taken for the surface of the Mg-Ti couple that was heat treated at 570 ⁰C. The Cu and Sn inter-layer is obvious to be homogenous with the structure of the couple. The atomic percentages for the elements found by EDX were 27.67% Mg, 71.64% Ti, 0.17% Cu and 0.06% Sn.

Figure 3. SEM micrograph and EDX spectrum for Mg-Ti couple that was heat treated at 570 oC. (can be seen in the manuscript)

Reviewer Comment: The corrosion rate could be directly compared with the pH value changes, mass loss, and hydrogen release, which are still strongly suggested, particularly when the authors think " it is expected that the pH values will decrease from the near neutral to the acidic values due to the dissolution of Mg".

Response to Reviewer: We performed the following experiments as recommended and as can be seen below (we could not measure the hydrogen release because all the experiments were carried out in open to air cell and also we don’t have the facility to measure it,

3.4. Weight-loss, pH, SEM images and EDX investigations

Figure 10. Variations of (a) weight-loss (∆W) and (b) corrosion rate (RCorr) as a function of time for (1) Mg AZ31 alloy, (2) Mg-Ti couple treated at 540 ⁰C, (3) Mg-Ti couple treated at 570 ⁰C, and (4) Mg-Ti couple treated at 600 ⁰C, in 3.5% NaCl solutions.

Figure 10 illustrates the variations (a) weight-loss (∆W) and (b) corrosion rate (RCorr) as a function of time for (1) Mg AZ31 alloy, (2) Mg-Ti couple treated at 540 ⁰C, (3) Mg-Ti couple treated at 570 ⁰C, and (4) Mg-Ti couple treated at 600 ⁰C, in 3.5% NaCl solutions. One can see easily that the value of ∆W increases and the values of RCorr decreases with the increase of time for all coupons. The increase of ∆W is due to the increase of dissolution of Mg and Mg-Ti couples with increasing the time of immersion. The increase of ∆W for AZ31 was higher than those recorded for Mg-Ti couples, which its loss in weight decreases with the increase of heat treatment temperature. On the other hand, the values of RCorr decrease due to the accumulation of corrosion products on the surface with increasing the time of immersion. It is also seen that the highest RCorr values were obtained for AZ31 alloy (curve 1) followed by the couples that were treated at lower temperatures, i.e., 540 ⁰C > 570 ⁰C > 600 ⁰C. The increase of weight-loss with time is due to the dissolution of the surface with time, while the decrease of corrosion rate comes from the accumulation of the corrosion products on the surface of the material leading to reducing the corrosion. This is in good agreement with the principles of the occurrence of uniform corrosion, which starts fast and decrease with time.

Figure 11. Variations of pH with time for (1) Mg AZ31 alloy, (2) Mg-Ti couple treated at 540 ⁰C, (3) Mg-Ti couple treated at 570 ⁰C, and (4) Mg-Ti couple treated at 600 ⁰C, in 3.5% NaCl solutions.

On the other hand, the change of pH values, which were obtained for the solutions during performing the weight-loss experiments for the different materials over 5 days immersion are shown in Figure 11. It is clear that the pH values initially increased from 6.22 (pH of the chloride solution without having materials immersed in) after 24 h of the immersion of the different materials for all samples and it was the highest for AZ31 alloy and decreased for the Mg-Ti couples, particularly with the increase of heat treatment temperature. The values of pH for all solutions were noticed to decrease with increasing immersion time, which is most probably due to the decrease of corrosion rate for our materials with time. This is also because the corrosion process is accompanied by a surface alkalinization that could lead to the initial increase of pH with time as reported in the earlier studies [26,31]. Therefore, the increase of corrosion rate due to the fast dissolution of materials after short immersion periods of time would lead to the increase of pH values, which would then decrease when the corrosion rate slows down due to the coverage of surface of with thick corrosion product layer. Furthermore, the increase of the heat treatment temperature led to further reduction in pH values as compared to the blank alloy (AZ31) that suffers more corrosion that could lead to raising the values of pH as well as the values RCorr Also, the increase of heat treatment temperature from 540 ⁰C to 600 ⁰C decreased the weight loss with time and therefore the values of RCorr. These weight-loss, RCorr and pH measurements are in good agreement with the electrochemical results that the corrosion of these materials decreased as per the following order; AZ31 > Mg-Ti couple treated at 540 ⁰C > Mg-Ti couple treated at 570 ⁰C > Mg-Ti couple treated at 600 ⁰C.

Reviewer Comment:  A more detailed corrosion mechanism should be included in the schematic diagram (Fig. 16).

Response to Reviewer: More details have been added to the text of the last paragraph in the Results and discussion part;

The SEM/EDX investigations thus confirm the data obtained by the electrochemical techniques and the mechanism of the corrosion process can be represented by the schematic diagram shown in Figure 19. Here, the dissolution of Mg takes place upon the increase of potential in the anodic direction as indicated from the polarization measurements and expressed by Eq. (3) releasing Mg2+. These cations react rapidly with hydroxyl groups presented in the solution to from Mg(OH)2, which is not protective enough to prevent the corrosion of Mg alloy and Mg-Ti couples in the chloride solution as depicted by Eq. (4). The formation of MgO also occurs on the surface of the test electrode as a result of the reaction between Mg and O (see Eq. (5)). However, the formation of this oxide does not protect the surface from being corroded under the aggressiveness action of the concentrated chloride solution.

Figure 19. Schematic diagram for the corrosion mechanism at the surface of Mg-Ti couples after being immersed in the chloride solution.

Round  3

Reviewer 1 Report

Accept.